# Perceptions of Drinking Water Service of the 'Off-Grid' Community in Cimahi, Indonesia



**Rizky Prayoga [1], Anindrya Nastiti [2,*], Seth Schindler [3], Siska W. D. Kusumah [2], Arief D. Sutadian [4], Eka J. Sundana [5], Elivas Simatupang [6], Arie Wibowo [7], Bagus Budiwantoro [8] and Majid Sedighi [9]**

[1] Study Program of Environmental Engineering, Faculty of Civil and Environmental Engineering, Institut Teknologi Bandung, Bandung 40132, Indonesia; rizkyprayoga@students.itb.ac.id

[2] Environmental Management Technology Research Group, Faculty of Civil and Environmental Engineering, Institut Teknologi Bandung, Bandung 40132, Indonesia; siskarius@gmail.com

[3] Global Development Institute, University of Manchester, Manchester M13 9PL, UK; seth.schindler@manchester.ac.uk

[4] West Java Research and Development Agency, Bandung 40286, Indonesia; ariefdhany@jabarprov.go.id

[5] West Java Development Planning Agency, Bandung 40135, Indonesia; eka.jatnika@jabarprov.go.id

[6] Cimahi City Development Planning Agency, Cimahi 40513, Indonesia; elivas@unpar.ac.id

[7] Material Science and Engineering Research Group, Faculty of Mechanical and Aerospace Engineering, Institut Teknologi Bandung, Bandung 40132, Indonesia; ariewibowo@material.itb.ac.id

[8] Mechanical Design Research Group, Faculty of Mechanical and Aerospace Engineering, Institut Teknologi Bandung, Bandung 40132, Indonesia; budiwan@edc.ms.itb.ac.id

[9] Department of Mechanical, Aerospace and Civil Engineering, University of Manchester, Manchester M13 9PL, UK; majid.sedighi@manchester.ac.uk

* Correspondence: anindrya@tl.itb.ac.id; Tel.: +62-22-2504952

**Abstract:** The establishment of decentralized drinking water systems in urban areas is technically and financially feasible, and these 'off-grid' systems can complement investment in traditional piped water systems. However, users often see 'off-grid' systems as the second-best option, compared to citywide piped water systems. Thus, although they are designed to improve access to water and reduce inequality, they can be perceived by users as infrastructural manifestations of extant inequality. In this paper, we present original research on the perceptions of users in Cimahi, Indonesia, surrounding their access to water and willingness to use and maintain 'off-grid' infrastructure. The majority of respondents used groundwater and packaged water as their primary water sources, and paid approximately twice the maximum tariff of piped water service. We interpreted the survey data with the theory of planned behavior framework and determined that respondents demonstrated a willingness to pay fees for 'off-grid' water systems, participate in water supply programs, and switch to new water sources. These intentions were affected by their attitude towards the behavior, subjective norm, and perceived behavioral control to various degrees. The findings are useful for those designing strategies to introduce novel water delivery systems aimed at improving water access for diverse and disadvantaged socioeconomic groups in urban areas in the Global South.

**Keywords:** Cimahi City; 'off-grid' communities; theory of planned behavior; water supply

## 1. Introduction

The provision of universal access to safe drinking water resources remains a major challenge [1,2]. As of 2017, more than 25% of the population of many countries in Sub-Saharan Africa, and South and Southeast Asia did not have access to improved water sources [3]. The lack of safe drinking water leads to increased incidence of water-borne diseases and results in significant opportunity costs related to work and school absences [4].

Challenges surrounding water provision are apparent in metropolitan areas across much of the Global South. Many cities have grown faster in terms of area and population than their infrastructure systems, undermining the 'modern infrastructural ideal' [5], in

which citywide infrastructure systems integrate households and provide near universal coverage. In the absence of centrally managed citywide systems, small-scale and 'off-grid' infrastructure systems have been introduced to meet the needs of residents. In some instances, these are public initiatives, while in other cases, they are presented by private-sector firms or incrementally auto-constructed by residents themselves and owned/operated by communities [6]. These decentralized and 'off-grid' solutions have attracted attention from scholars, policymakers, and residents because it is doubtful that the target of universal access will be met solely using centralized piped water systems [6]. In addition to the high cost of expanding centralized piped water systems [7], the breakdown of ageing infrastructure has raised questions surrounding the cost-effectiveness and technical viability of such systems [8]. The proliferation of 'off-grid' infrastructure initiatives has resulted in the emergence of water systems in many southern metropolises as a disconnected patchwork of sub-systems that vary in terms of quality, efficiency, cost, and accessibility, and comprise unique heterogeneous infrastructure configurations [9–12].

Decentralized water provision aims to increase local responsiveness, and target vulnerable groups and reduce inequality. One expectation is that, if given a choice, people will opt for low-cost sustainable alternatives to piped water, and that they may even contribute to the maintenance of these systems [13]. However, monitoring trends show that, over time, decentralized water supply services consistently fail to function as envisioned [14]. For example, up to 70% of rural water schemes in Sub-Saharan African countries are projected to be nonfunctional or intermittently functional at any given time [14]. The malfunctions are attributed to the lack of a sense of ownership from the community, the failure of providers to understand the local context, and the low willingness of users to pay [15,16]. This research seeks to assess the willingness of residents to pay for water from a decentralized 'off-grid' system and participate in its maintenance.

This paper explores the sociotechnical interface in an urban community in *Kelurahan* Citeureup, Cimahi City, a suburb of the Bandung Metropolitan area. Like most large Indonesian cities, the Bandung Metropolitan area exhibits a heterogeneous infrastructure configuration. Indeed, Indonesia is the fourth most populous country in the world, and only 9.87% of the population enjoys access to piped water [17]. Most people are forced to rely on groundwater to meet their water needs [17]. In addition to posing significant health risks, groundwater extraction is not sustainable due to its low recharge rate and the uneven geological distribution of aquifers [18]. The achievement of universal access to piped water by monopolistic providers in Indonesian cities under the existing water supply policy is unlikely [19]. Thus, rising water demands require innovative supply-oriented solutions that are economically viable and ecologically sustainable. A household survey was conducted (by telephone due to the COVID-19 pandemic, which inhibited face-to-face research), the objective of which was to assess respondents' willingness to pay for 'off-grid' water provision and participate in the maintenance of the system. The research also sought to determine the relationship between respondents' knowledge and attitudes towards 'off-grid' water systems and their knowledge surrounding water pollution. The responses were analyzed with the theory of planned behavior, or TPB [20]. We determined that the fundamental factors that influence water usage are perceptions regarding costs and benefits, culture, health and hygiene values/practices, and environmental sustainability [8]. This research contributes to scholarship on 'off-grid' water systems that have primarily focused on rural areas [21,22], by examining the sociotechnical interface in a fast-growing urban area in the Global South.

In the following two sections, we introduce the conceptual and analytical frameworks, as well as the research site. In section four, we introduce the profile of respondents, and in section five, we present findings and analysis. We discuss the results in section six, and conclude in section seven.

## 2. Framework

To explain the factors that influence specific behaviors, a variety of theoretical frameworks have been established. Ajzen and Fishbein's theory of reasoned action has served as the foundation for more advanced models and structures to describe the attitude–behavior relationship [23]. This research focused on the theory of planned behavior (TPB) as an extension of the theory of reasoned action (Figure 1). According to TPB, behaviors are predicted by behavioral intentions (I), which are predicted by attitude, subjective norms, and perceived behavioral control. A general attitude toward behavior (ATB) is one of the attitudinal factors; individuals' perceptions of norms and conventions about a specific action, i.e., how others will perceive that behavior and motivation to conform to certain views, are referred to as subjective norms (SN); meanwhile, individuals' perceptions of how simple or difficult it is to execute an action are referred to as perceived behavior control (PBC) [24].

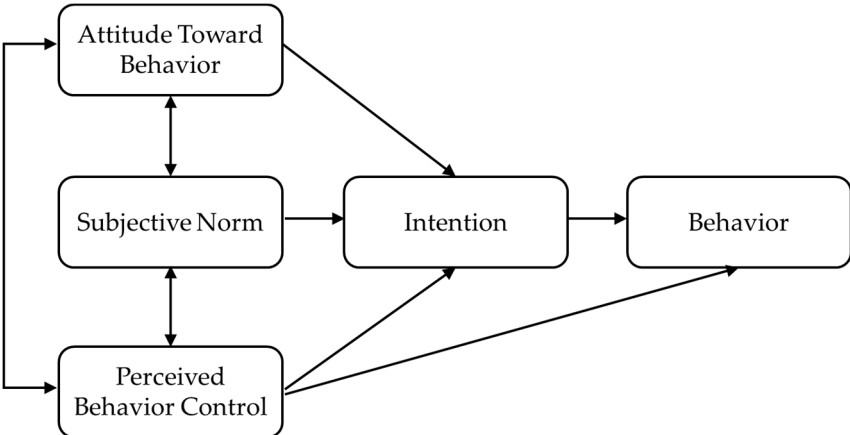

**Figure 1.** Theory of planned behavior (TPB) framework, adapted from [20].

Even though various theoretical and practical issues exist in the use of the TPB, the advantages outweigh the disadvantages. TPB is more effective compared to non-theory-based interventions [25], an effective framework for guiding the design of a behavior-change intervention [26], widely used in many different studies, provides reliable, valid measures, and fits with statistical modellings [20]. TPB has been successfully applied in attempting to predict change in behavior in different research areas, such as in health [27], education [28], energy [29], food [30], tourism [31], and agriculture [32]. TPB has been utilized in numerous studies related to pro-environmental behaviors (e.g., [33,34]). Thus, TPB is an appropriate tool that can be used to answer the central questions of this study.

## 3. Materials and Methods

### 3.1. Cimahi City

With an area of 40 km$^2$, the population and population density of Cimahi City is around 585,860 people and 13,775 people/km$^2$, respectively. Cimahi City consists of three subdistricts and 15 urban villages. Figure 2 shows Cimahi City, Indonesia. Rapid population growth has led to increased demand for water, causing water to become a prominent social and economic issue. Until 2017, piped water served 20.67% of the total population of Cimahi City, while the rest of the population relied on several sources, such as groundwater and water from vendors [35]. In this study, the specific research location was Citeureup, an urban village located in the North Cimahi District with an area of 323,535 ha. There were 12,902 households within the population of Citeureup in 2020, reaching 37,433 people and a population density of 116 people/ha [36]. This specific location was selected since most households still depend on groundwater as a water source. As noted above, this leads to environmental degradation, and there has been evidence of

land subsidence. To date, 88% of the residents in Citeureup collect at least some water from non-piped sources (bottled, well, and spring water) [37].

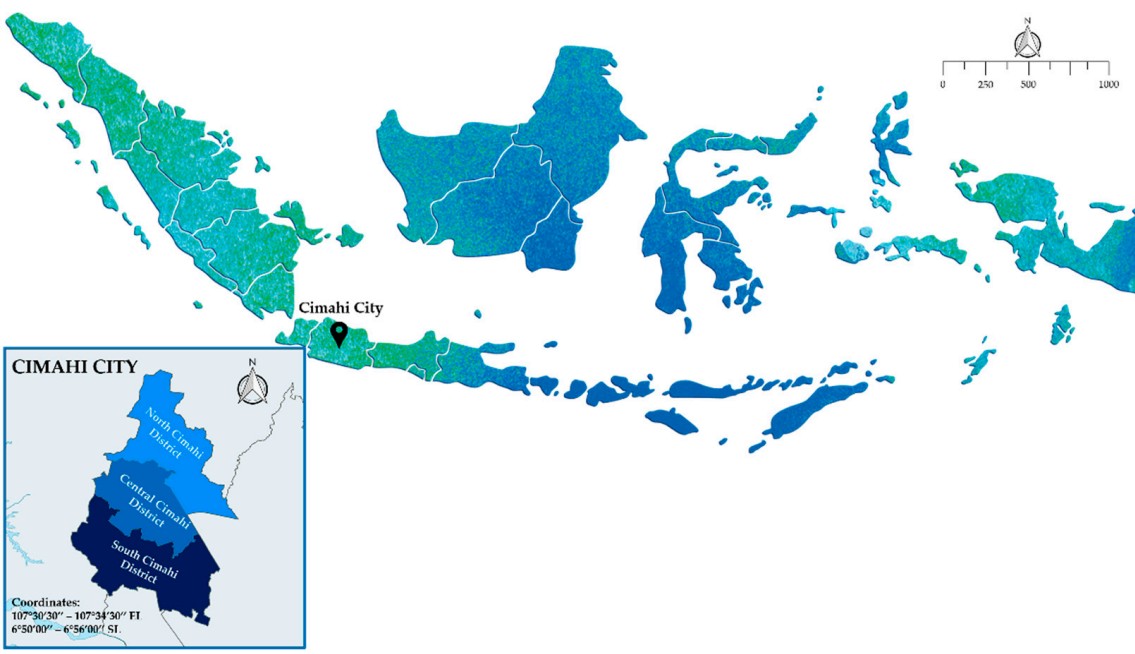

**Figure 2.** Cimahi City, Indonesia.

### 3.2. Cross-Sectional Survey

To understand the perception and behavior of people in Citeureup, Cimahi, we conducted a cross-sectional telephone survey. A sample of 100 Citeureup residents, recruited through convenience sampling [38] with $e$=10%, participated in the survey. All respondents were those who were not served by piped water service. The survey design was informed by the TPB [20], and consisted of attributes of respondents, the condition of drinking water supply, the perception of the respondents on the quality, quantity, and continuity of the existing drinking water supply system, knowledge of the water supply system, and the perceptions of the drinking water supply system. Prior to distributing the questionnaire, it was tested in a pilot survey to ensure the questions were clear to respondents. After this pilot survey, the questionnaire was refined. The key items in the questionnaire are shown in the Appendix A, while the full questionnaire is shown in Supplementary Material. The research framework is shown in Figure 3.

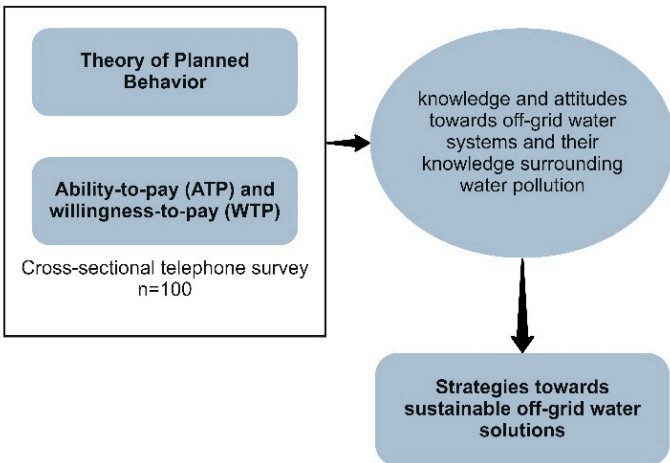

**Figure 3.** Research framework.

### 3.3. Analysis

Scores were then assigned to respondents' answers. In questions regarding knowledge of water supply, a correct answer elicited a score of 1, while a wrong answer elicited a score of 0. Respondents were then classified into *low knowledge* (score between 0 and 6) and *high knowledge* (score between 7 and 12). Meanwhile, in questions related with the components of TPB, "strongly agree" elicited a score of 5, "agree" 4, "neutral" 3, "disagree" 2, and "strongly disagree" elicited a score of 1. To obtain the average score of each TPB component for further statistical analysis, the respondents' responses were averaged. Spearman rank analysis was selected in order to understand the correlation between variables [39]. Moreover, a multiple regression analysis was conducted to understand the link between the component of TPB and I.

Respondents' ability-to-pay was measured by analyzing water expenditure and income [40]:

$$ATP = \frac{I_t \times P_p}{T_t} \tag{1}$$

Note:

ATP: Ability-to-pay (per m$^3$ of water)
$I_t$: Total household income per month
$P_p$: % of water expenditure from non-food expenditure per month
$T_t$ : Total household water use per month, m$^3$/month

Total household income per month ($I_t$) was estimated based on the structured interview using a questionnaire. The value of $P_p$ was calculated by referring to the percentage of Cimahi City's non-food expenditure based on *West Java Province in Figures 2021* and the maximum water expenditure value, which refers to the *Regulation of the Minister of Internal Affairs Number 71 of 2016*. The percentage of Cimahi City residents' non-food expenditure in 2020 was 56.52% [41], and the maximum expenditure value for water is 4% [42]. The value of $P_p$ can be calculated using the following equation [43]:

$$P_p = \% \, Non - food \, expenditure \, \times \, \% \, Maximum \, water \, expenditure \tag{2}$$

$$P_p = 56.52\% \, \times \, 4\% = 2.26\% \tag{3}$$

Based on data from the Cimahi City Spatial Plan (RTRW) 2012–2017, water consumption in Citeureup Village is 115 L/capita/day (0.115 m$^3$/c/d) [44]. The number of people in one household in the West Java Province is 3.8 ($\approx$4) [41]. $T_t$ can be calculated using the following equation:

$$T_t = Water \, use \, \left(m^3/c/d\right) \, \times \, Number \, of \, people \, in \, household(c) \, \times \, 1 \, month(d) \tag{4}$$

$$T_t = 0.0115 \, m^3/c/d \, \times \, 4 \, capita \, \times \, 30 \, day = 13.8 \, m^3 \tag{5}$$

In addition, the people of Citeureup spend a certain amount of money to meet their water needs. Water expenditure may not match the calculated ability-to-pay (ATP) value. To be able to find out the cost of water (per m$^3$) incurred by the respondent, the following equation can be used:

$$Water \, cost \, \left(per \, m^3\right) = \frac{Total \, water \, expenditure \, (IDR)}{T_t} \tag{6}$$

Moreover, respondents' willingness-to-pay was measured using a contingent valuation, in particular, stated preference method [45]. We asked respondents directly about their WTP: How much is your household willing to pay for a quality water supply service? The WTP value of the Citeureup was determined using the bidding game method [46]. This method sorts the WTP class from the community and shows the frequency of the number of respondents who filled in the WTP value according to the existing WTP class.

After that, the following formula can be used to calculate the WTP value of the Citeureup community [46]:

$$EWTP = \sum W_i \times P_{fi} \tag{7}$$

Note:

EWTP: Estimated value of WTP
$W_i$: Value of WTP *i*
$P_{fi}$: Relative frequency

All respondents were free to withdraw from the interview anytime. Ethical approval was obtained from the Research Ethic Committee of Universitas Padjajaran, with the number 32/UN6.KEP/EC/2021.

## 4. Respondent Profile

Table 1 shows the profile of respondents, based on the structured interview using a questionnaire. The $R^2$ value obtained shows that education affects income by 15.52%. The correlation coefficient (r) between education and income is 0.394 (sufficient correlation). The positive value of r shows that the relationship of the two variables is unidirectional, where the higher the education, the higher the income earned.

**Table 1.** Respondents' profile.

| Attribute | Responses |
| --- | --- |
| Gender | Male = 56%; Female = 44% |
| Age | <20 = 14%; 21–30 = 27%; 31–40 = 17%; 41–50 = 20%; 51–60 = 19%; 61–70 = 3% |
| Education | Primary school = 3%; junior high school = 11%; high school = 47%; diploma = 10%; undergraduate = 25%; postgraduate = 4% |
| Income | Low (<IDR 3,241,929 or <USD 233.17) = 45%; Middle (IDR 3,421,930–IDR 6,483,858 or USD 223.17–USD 471.12) = 32%; High (>IDR 5,483,858 or > USD 471.12) = 23% |
| Status | Head of household = 41%; Household member = 59% |

## 5. Results

### 5.1. Access, Quality, Quantity, Continuity, and Affordability of Water Supply

Figure 4 shows the source of water used by respondents for drinking (a), and bathing and washing (b). It was demonstrated that groundwater, and branded bottled water and refill water are the primary sources for drinking. In Indonesian cities, there are two types of bottled water: branded bottled water and non-branded bottled water purchased from refill water kiosks, which is cheaper than their branded counterpart, popularly known as 'refill water'. The refill water kiosks are an alternative option for low-income households in urban areas without piped water supply service [47]. Meanwhile, groundwater extracted through boreholes is the primary source for cooking, bathing, and washing. In Cimahi, the data show that people often use more than one water source.

We also asked the perceptions of the quality of 'service' of the water supply in terms of quality, quantity, continuity, and affordability. As many as 88% of the respondents thought the water that they use is of good quality because the water is colorless and the water has no taste or smell. In terms of quantity, 68% of respondents feel that the quantity of water they have meets their water needs, and 32% of respondents feel it does not meet their water needs. As a result, these respondents source alternative water supply by buying water, asking the neighbors, and taking water from the nearby mosques. In terms of continuity, 83% of respondents have water available for 24 h, and 17% do not have water available for 24 h.

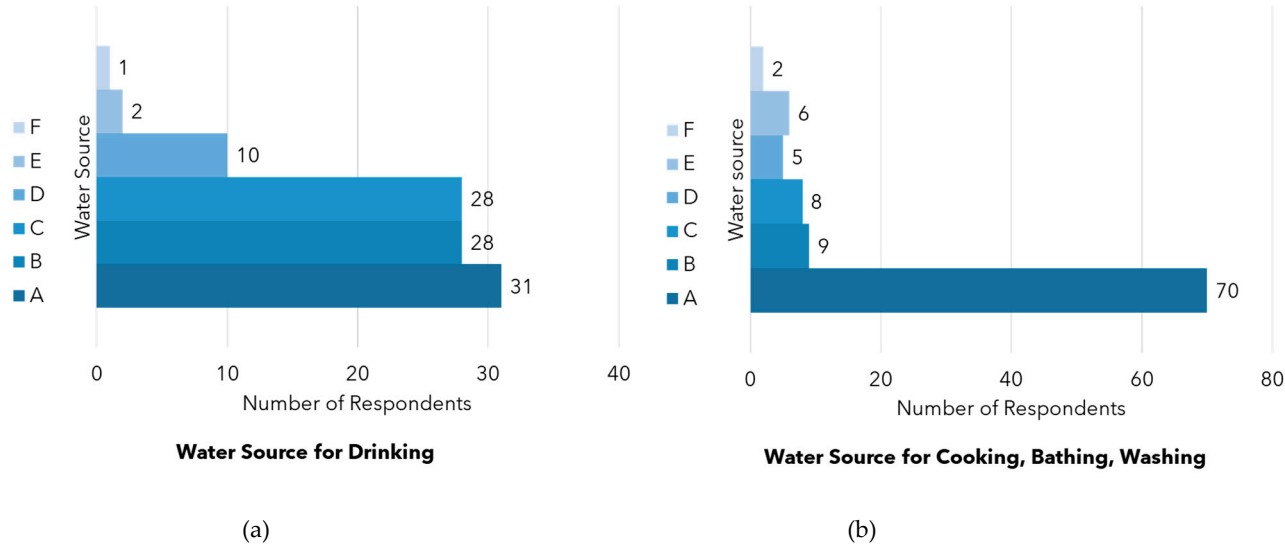

(a)                                                                  (b)

**Figure 4.** (**a**) Water sources for drinking, where: A. borehole or dug well; B. refill kiosks; C. bottled water; D. bottled water and refill kiosks; E. borehole or dug wells and bottled water; F. borehole or dug well and refill kiosks; (**b**) Water sources for cooking, bathing, and washing, where: A. borehole; B. dug well; C. borehole and bottled water; D. dug well, bottled water, and refill water; E. boreholes and dug well; F. dug well and water vendors; G. borehole and refill kiosks; H. dug well and refill kiosks.

Figure 5 shows public opinions regarding water affordability. The majority (61%) of the respondents spend IDR 10,000 (USD 0.7) to IDR 100,000 (USD 6.97) for water. Based on income, water expenditure for low-income households is IDR 77,556 (USD 5.34). Meanwhile, water expenditure for middle-income and high-income households are IDR 134,375 (USD 9.25) and IDR 182,391 (USD 12.56), respectively. The average expenditure of water is IDR 119,850 (USD 8.3). With the assumption that total water demand per month is 115 L per capita per day, and the total number of persons in a household is four persons, the average unit price for water among 'off-grid' households is IDR 8,685/m$^3$ or USD 0.6/m$^3$. This figure is much higher compared to the tariff of piped water service in Cimahi City that varies between IDR 1,800 (USD 0.13) and IDR 5,000 (USD 0.34) per m$^3$.

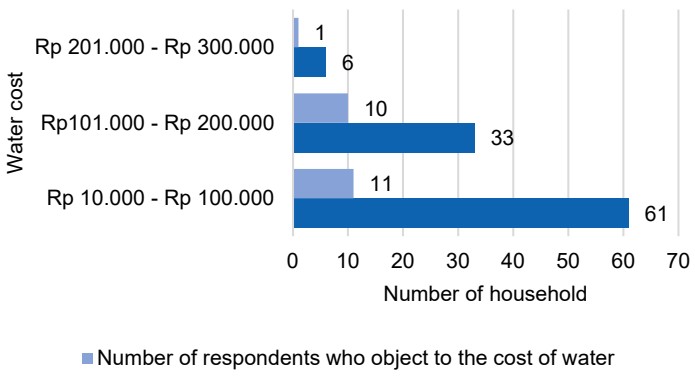

■ Number of respondents who object to the cost of water

■ Number of respondents paying for water

**Figure 5.** Public opinion regarding water affordability. The dark blue represents respondents who pay for water within a certain range, while the light blue represents respondents who feel they pay too much for water.

We also calculated the proportion of water expenditure with income. It was found that water expenditure for low-income, middle-income, and high-income households is 3.4, 2.69, and 1.64% from their income, respectively. Furthermore, we calculated the households' ATP and compared it to the actual expenditure (per m$^3$) by income (Table 2).

**Table 2.** The number of households whose water expenditure exceeds their ATP.

| Income | Number of Households, % |
|--------|-------------------------|
| Low | 40 out of 45 households, 88.9 |
| Middle | 24 out of 32 households, 75 |
| High | 2 out of 23 households, 8.6 |

Although poor households pay the lowest in terms of absolute water expenditure, they bear the highest burden of water costs, reflected by the proportion of water expenditure to income. This burden is also expressed in terms of the percentage of households paying more than their ATP per $m^3$.

Based on the WTP survey, low-income households are willing to pay IDR 59,002 (USD 4.06), middle-income households are willing to pay IDR 98,375 (USD 6.77), and high-income households are willing to pay IDR 132,652 (USD 9.13) per month. In line with a study in Nepal [48], we saw that, as expected, WTP increases as income rises. The WTP in Citeureup, Cimahi, is also higher compared to the tariff of piped water services in Cimahi City (IDR 51,060 or USD 3.54 per month for a four-person household). On average, WTP for all households is lower than their actual water expenditure but higher than the existing piped water tariff.

*5.2. Knowledge and Perception of the 'Off-Grid' Community*

Figure 6 shows the knowledge of the respondents based on income level. The survey suggests that low-income respondents have less knowledge surrounding infrastructure and water quality, compared to middle-income and high-income respondents. This is indicated by the percentage of people with low knowledge regarding water supply decreasing as the income increases. A Spearman rank analysis also suggested a strong correlation between income and knowledge on water supply (r = 0.528). Moreover, the positive value of the correlation coefficient shows that the relationship between the two variables is unidirectional, where the higher the income, the higher the knowledge related to water supply. The $R^2$ value obtained shows that income influences knowledge by 27.87%. Furthermore, we also correlated education and knowledge. The r value of 0.368 suggested a sufficient correlation between education and knowledge. Education affects knowledge by 13.54%.

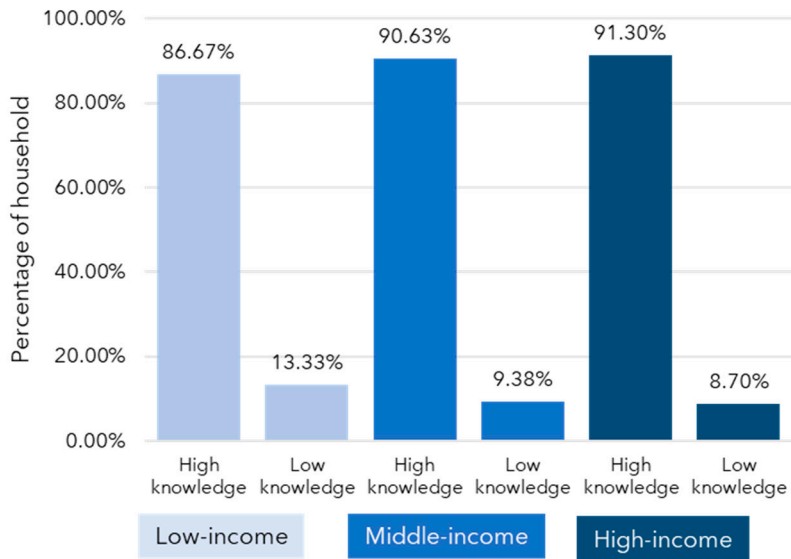

**Figure 6.** Knowledge on water supply by income level.

Table 3 shows the average score of attitudes towards behavior (ATB), subjective norm (SN), perceived behavioral control (PBC), and intention (I) based on income level.

**Table 3.** Average score of the components of TPB (scale 1–5).

| Income | ATB | SN | PBC | I |
|--------|-----|-----|-----|-----|
| Low | 4.11 | 3.49 | 4.08 | 4.02 |
| Middle | 4.23 | 3.55 | 4.10 | 3.96 |
| High | 4.09 | 3.77 | 4.08 | 4.09 |

ATB refers to the degree to which a respondent has a positive or negative assessment of aspects related with drinking water supply. It was shown that the middle-income respondents have the most positive evaluations in terms of willingness to participate in drinking water programs. They also strongly believe that the solid waste problems will affect the quality of water, which in turn will affect human health. These respondents perceived that good quality water costs money to obtain, and that community participation is needed in drinking water programs. SNs are the general societal pressures that a respondent feels to practice or not practice certain behaviors, such as, in this case, disposing of solid waste in the designated place, paying water fees, participating in water supply programs, and switching to new water sources. SN score is the highest among high-income households. The influence of the surrounding community, such as neighbors and community leaders, on the perceptions of the high-income community is the strongest. PCB refers to the perceived capacity to engage in a specific behavior, in this case, disposing of solid waste in the designated place, paying water fees, participating in water supply programs, and switching to new water sources. PCB is the highest among middle-income households. Meanwhile, high-income households have the highest intentions to dispose of solid waste to designated behavior, pay water fees, participate in the new water systems, and switch to new water sources.

Using a multiple linear regression analysis, we then analyzed the effect of ATB, SN, and PBC towards intention (I) among low-income, middle-income, and high-income respondents. The result is shown in Table 4.

**Table 4.** The effect of ATB, SN, and PBC towards I among low-, middle-, and high-income households.

| Variable | R | $R^2$ | Reg. Coeff | Constant | SE | SR |
|----------|-----|-----|-----|-----|-----|-----|
| **Low-Income Household** | | | | | | |
| ATB | | | 0.422 | | 15.92% | 49.59% |
| SN | 0.566 | 0.320 | 0.202 | 0.684 | 8.63% | 26.89% |
| PBC | | | 0.219 | | 7.55% | 23.52% |
| **Middle-Income Household** | | | | | | |
| ATB | | | 0.565 | | 22.69% | 45.28% |
| SN | 0.708 | 0.501 | 0.217 | −0.267 | 13.52% | 26.99% |
| PBC | | | 0.285 | | 13.90% | 27.74% |
| **High-Income Household** | | | | | | |
| ATB | | | 0.120 | | 7.30% | 10.80% |
| SN | 0.822 | 0.676 | −0.163 | 1.167 | −4.95% | −7.32% |
| PBC | | | 0.745 | | 65.24% | 96.52% |

From the results of multiple linear regression analysis, the intention equation was obtained as follows:

$$I = 0.422ATB + 0.202SN + 0.219PBC + 0.684 \text{ for low-income households}$$
$$I = 0.565ATB + 0.217SN + 0.285PBC − 0.267 \text{ for middle-income households}$$
$$I = 0.120ATB − 0.163SN + 0.745PBC + 1.167 \text{ for high-income households}$$

From Table 4, R indicates the relationships between the three components of TPB and I simultaneously. For example, among low-income households, there is a strong relationship between the three variables of TPB to intention (R=0.566). $R^2$ indicates to what extent all TPB components contribute to I. For example, among low-income households, the $R^2$ value of 0.320 means that the three variables in this study influence 32% on intention. There are 68% other variables that affect the intention of low-income people that are not included in the scope of this study. Meanwhile, effective contribution (SE) is a measure of the contribution given by each independent variable to the dependent variable. The sum of each effective contribution in each independent variable has the same value as the determinant coefficient ($R^2$) in multiple linear regression analysis. Lastly, relative contribution (SR) is a measure that explains the size of the contribution of an independent variable to the $R^2$ value. The total number of relative contributions from all independent variables is 100%. For example, among low-income households, the ATB variable has an effective contribution on the intention by 15.92%, or equivalent to 49.59%. The SN variable gives contribution to intention effectively by 8.63%, or equal to 26.89% relative effect. The PBC variable has an effective contribution on the intention of 7.55%, or equivalent to 23.52% relative contribution.

Table 4 also suggested that the relationship between TPB components and I (depicted by R), and the influence of TPB components to I (depicted by $R^2$) increase as income rises. The positive R and $R^2$ values indicate that higher ATB, SN, and PBC leads towards higher I. It was also suggested that, among low-income and middle-income households, ATB provides the highest contribution to I. Meanwhile, among high-income households, PBC provides the highest contribution to I.

## 6. Discussion

Our objective was to better understand perceptions regarding water supply systems, and determine the willingness of residents to pay for, use, and maintain 'off-grid' water systems. Our result shows that the vast majority use groundwater extracted from boreholes, and packaged water. People in the Global South commonly use secondary water sources when the primary water source is perceived to fail in delivering water with the expected level of service quality [49,50]. To improve water quality, the community takes several measures, such as applying alum and using filters. These methods are two low-cost options with acceptable efficacy in removing microbial contaminations in water [51].

The most significant finding is that there is evidence that low-income community members would be willing to pay for, use, and maintain a cost-effective 'off-grid' water supply system. Our findings also confirmed that low-income respondents bear the highest burden of water costs, reflected from the proportion of water expenditure to income and the percentage of households paying more than their ATP. These trends are in line with a study that mentions that the proportion of household expenses devoted to water is inversely proportional to household income [52]. Except for low-income households, households' ATP in Citeureup is still within the World Bank's suggested range of international water affordability benchmarks of 3–5 percent of household income [53]. Studies in Asian and African countries have suggested that water is most costly for those who are not linked to public piped water systems [54,55]. Low-income residents are willing to embrace 'off-grid' infrastructure, but it must compete with groundwater and packaged water in terms of cost. We estimated that WTP for all households is lower than their actual water expenditure, but higher than the existing piped water tariff. This suggests that any new water systems in Cimahi should set the tariff based on the WTP calculated in this study. There is scope to increase tariffs for middle- and high-income households as their WTP is significantly higher than the existing water tariff. The WTP, however, may be overestimated as it answers the hypothetical contingent valuation questions [48].

More important than cost, respondents had limited knowledge of water quality, and they considered water suitable for drinking if it was colorless and odorless, although they exhibited anxiety about contaminated water (particularly from improperly disposed of

waste). This signals a need for education surrounding water safety, which could increase willingness to pay for treated water. Moreover, information regarding water quality must be readily available [56,57] so households can base their decision on the most current water quality information.

This study found that knowledge is related to income and education: the higher the income and education, the higher the knowledge surrounding water supply. This pattern has been explained in research explaining the effect of income on environmental knowledge [58]. Income has been shown to be positively related to education, and also positively related to environmental awareness [58]. Moreover, perceptions were measured using components of TPB. We found that among low- and middle-income households, interventions can be focused on providing a positive experience and increasing knowledge to further strengthen ATB, which has the highest contribution to I. Using multiple linear regression analysis, we also found that the higher the attitude towards behavior, subjective norms, and perceived behavioral control scores, the more significant intention. This trend is in line with a study in Bulgaria [59], where the TPB variables showed a positive correlation with intention.

In light of our findings, 'off-grid' solutions are designed as low-cost systems that have the potential to reduce inequality and provide vulnerable communities with access to water. Still, users may interpret them as poor substitutes for centralized systems that have characterized modernist urban development and planning. If public perception is the primary barrier to widespread adoption of 'off-grid' water systems, public awareness campaigns may sensitize potential users. However, if 'off-grid' systems are not perceived as second-best, then their limited success can be attributed to other factors (e.g., durability or organizational structures that do not welcome community participation).

## 7. Conclusions

This research examined the willingness of residents to embrace 'off-grid' water infrastructure by focusing on the sociotechnical interface in a peri-urban community in Indonesia. This research is important because 'off-grid' systems have the potential to complement existing piped water systems, but they have met with mixed success. As noted, many pilot projects fell into disrepair. We hypothesized that one reason might be a lack of enthusiasm among users, who view 'off-grid' sourced water unfavorably in comparison to water obtained from centralized piped water systems. Our research shows that Cimahi residents were not opposed to expanding 'off-grid' systems in principle. This is significant because the provision of affordable and safe drinking water in Cimahi has never been more urgent. Indeed, this research was conducted by telephone, given limitations on face-to-face research imposed by COVID-19. The pandemic has highlighted the importance of resilient water supply systems, and 'off-grid' systems have the advantage that they can be rolled out quickly and supply can be scaled up in times of health or environmental crises. Thus, they can complement existing piped water systems, and we conclude that they should be more widely piloted in urban areas in developing countries to determine the conditions under which they enhance access and reduce inequality.

**Supplementary Materials:** The following are available online at https://www.mdpi.com/article/10.3390/w13101398/s1, Table S1. Questionnaire: Perceptions of off-gird community regarding drinking water supply systems in Cimahi City.

**Author Contributions:** Conceptualization, A.N. and S.S.; methodology, A.N., R.P., and S.W.D.K.; validation, A.D.S., E.J.S., and E.S.; formal analysis, R.P. and A.N.; investigation, R.P. and S.W.D.K.; resources, A.W., B.B., and M.S.; data curation, R.P.; writing—original draft preparation, R.P., A.N. A.D.S., E.J.S., and E.S.; writing—review and editing, A.N. and S.S; visualization, R.P., A.W., and A.N.; supervision, S.S.; project administration, A.W., B.B., and M.S.; funding acquisition, A.W., B.B., M.S., and S.S. All authors have read and agreed to the published version of the manuscript.

**Funding:** This research was funded by Newton Fund *Institutional Links Grant* entitled "Socio-technical solutions to water security challenges in urban areas and post-disaster scenarios" (pfact 71290) and Ke-

menterian Riset dan Teknologi/Badan Riset dan Inovasi Nasional (Number 2/E1/KP.PTNBH/2020 and Number 2/E1/KP.PTNBH/2021).

**Institutional Review Board Statement:** The study was conducted according to the guidelines of the Declaration of Helsinki, and approved by the Research Ethic Committee Universitas Padjajaran, Indonesia (Ethical Approval Number 32/UN.6.KEP/EC/2021, dated 13 January 2021).

**Informed Consent Statement:** Informed consent was obtained from all subjects involved in the study.

**Data Availability Statement:** The data presented in this study are available on request from the corresponding author. The data are not publicly available due to privacy issue.

**Acknowledgments:** We thank all respondents. We also thank Rivan Dachlan Tanzah for providing administrative support in this paper. We also thank three anonymous reviewers for their comments.

**Conflicts of Interest:** The authors declare no conflict of interest. The funder had no role in the design of the study; in the collection, analyses, or interpretation of data; in the writing of the manuscript, or in the decision to publish the results.

## Appendix A

**Table A1.** Key questions related to knowledge and components of the TPB framework.

| Questionnaire Items | Category |
|---|---|
| Drinking water can be consumed without treatment | Knowledge |
| Water quality must be tested first and meet the health requirements before it can be consumed | Knowledge |
| Water that is colorless, odorless, and tasteless has good quality | Knowledge |
| The quality of water used to meet daily needs does not affect human health | Knowledge |
| Various water sources available in nature have begun to deteriorate due to environmental pollution | Knowledge |
| The use of groundwater with boreholes/artesian wells to meet water needs can damage the environment | Knowledge |
| All Indonesian people have the same right to get water in sufficient quantities and of good quality | Knowledge |
| The government is obliged to provide drinking water for all Indonesian people | Knowledge |
| Waste that is disposed of in water bodies does not affect the production process carried out by the municipal water company | Knowledge |
| The community can receive water access from the government for free without being charged a fee | Knowledge |
| A quality drinking water supply system can be implemented if the community actively participates in the system | Knowledge |
| I believe that water quality will affect human health | ATB |
| I believe that I must participate in the drinking water supply system to meet my daily water needs (bathing, washing, cooking, drinking, etc.) | ATB |
| I believe that the solid waste problem will affect the quality of the water I use to meet my daily water needs (bathing, washing, cooking, drinking, etc.) | ATB |
| I believe that to get good quality water in sufficient quantities to meet daily needs (bathing, washing, cooking, drinking, etc.) costs money. | ATB |
| I believe community participation is needed in drinking water programs | ATB |
| I believe that new water service from community-based supply is superior in term of quality/quantity/affordability compared to my existing water sources. | ATB |

**Table A1.** *Cont.*

| Questionnaire Items | Category |
|---|---|
| I will dispose of solid waste in its designated place if people around me (family, neighbors, friends) do the same | SN |
| I will dispose of solid waste in its designated place if I get advice from community leaders (religious leaders, heads of neighborhood, etc.) | SN |
| I will pay the water fees even though the price increases if the people around me (family, neighbors, and friends) do the same. | SN |
| I will pay water fees even though the price increases if I get advice from local community leaders (religious leaders, heads of neighborhood, etc.) | SN |
| I will participate in water supply program if the people around me (family, neighbors, and friends) do the same. | SN |
| I will participate in water supply program if I get advice from local community leaders (religious leaders, heads of neighborhood, etc.) | SN |
| I will switch to a new water supply service from the community-based supply if the people around me (family, neighbors, and friends) do the same. | SN |
| I will switch to a new water supply service from the community-based supply if I get advice from local community leaders (religious leaders, heads of neighborhood, etc.) | SN |
| I can provide waste containers at my house to dispose of solid waste in its designated place | PBC |
| I will dispose of solid waste in its designated place if there are officers who take away solid waste from my house regularly | PBC |
| I have enough time to participate in water supply programs | PBC |
| I have enough money to participate in water supply programs | PBC |
| I will switch to a new water supply service from the community-based supply if the quality/quantity were acceptable or if it were affordable | PBC |
| I will not throw garbage or waste into water bodies | I |
| I will pay water retribution even though the tariff increases | I |
| I will participate in water supply program, in-kind or in cash | I |
| I will switch to a new water supply service from the community-based supply if such sources were available | I |

ATB = attitude towards behavior; SN = subjective norms; PBC = perceived behavioral control; I = intention. The options for knowledge questions are "right" and "wrong", while the options for TPB questions are "strongly agree", "agree", "neutral", "disagree", and "strongly disagree".

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
