# Peer review of "Perceptions of Drinking Water Service of the ‘Off-Grid’ Community in Cimahi, Indonesia"

_water, doi:10.3390/w13101398_

Round 1
Reviewer 1 Report
The abstract is correctly formulated, indicates background of study, main objective, some findings and conclusions.
Introduction should be rewritten.
Authors should be aware that the scientific papers have the common structure, and each section should follow the requirements. This Introduction should be rewritten, according to the structure which should be included in the research papers. The introduction of a research paper should contain a few other parts/ elements such as the chief goal(s) and objectives of the research, a brief but informative outline of the following content, explained, concept definitions, a brief history of the research into the topic, recent related discoveries, etc.). I suggest to rewrite the research questions and convert that to sub-objectives of the study. Please be aware that “Water MDPI” is technical journal.
Please provide the context of novelty in this research.
- Framework &3. Materials and Methods:
Please improve the quality of Figure 1, Figure 2.
Usually the research questions are provided as the supplementary materials.
Sources of data for each equation is needed.
It will be good to provide the scheme of the research framework.
Results:
Please add main section results, and indicate sections 5 & 6 as sub-sections.
Discussion:
There is lack of this section. Please provide the comparison with some other paper evaluating the public acceptance. For now there is lack of deep discussion of the obtained results. Some information provided in conclusions could be moved to Discussion.
Conclusions:
Please provide the conclusions, not repetition of the results.
References:
Should be adapted to journal template.
Overall:
The article is well written. Authors put a lot of work to conduct the research an present the results. The topic is worth to be investigated.
There are some grammatical mistakes, please verify text.
Reviewer 2 Report
Figure 2 is hazy, kindly re-draw in good ways with coordinate
Kindly shift the Table 1 in to the supplementary section
Kindly indicate the y-axis of the figure 4, 5
Line no: 390-392: need support (kindly see: Long term trend analysis and suitability of water quality of River Ganga at Himalayan hills of Uttarakhand, India, Environmental Technology & Innovation 22, 101405; Quality assessment of springs for drinking water in the Himalaya of South Kashmir, India. Environ Sci Pollut Res 28, 2279–2300 (2021). https://doi.org/10.1007/s11356-020-10513-9)
Kindly check the sub-script and super-script throughly (e.g., Line number 346: R2 value)
Conclusion need to be re-written in technical way instead of generalized
Reviewer 3 Report
LINES 389-397 (version 2): linear regression analysis
I suggest to change
"Y" by "I"
"X1" by "ATB"
"X2" by "SN"
"X3" by "PBC"
As those are the acronyms used all over the manuscript
In fact, lines 393 to 397 could be deleted
Author Response
Dear Reviewer,
We are grateful for your remarks so this paper can be improved. We have changed "Y" by "I", "X1" by "ATB" "X2" by "SN", and "X3" by "PBC". We have also deleted lines 393 to 397. Thank you for your suggestion.
Round 2
Reviewer 1 Report
Article was sufficiently corrected.
Author Response
Dear Reviewer,
We are grateful for your remarks and suggestions so this paper can be greatly improved.
Reviewer 2 Report
All the best for upcoming research work
Author Response

(The authors gave the same response as above.)
